# Task-oriented machine learning surrogates for tipping points of agent-based models

Gianluca Fabiani [1,2], Nikolaos Evangelou[2], Tianqi Cui[2], Juan M. Bello-Rivas[3], Cristina P. Martin-Linares[4], Constantinos Siettos [5] & Ioannis G. Kevrekidis [2,3,6]

We present a machine learning framework bridging manifold learning, neural networks, Gaussian processes, and Equation-Free multiscale approach, for the construction of different types of effective reduced order models from detailed agent-based simulators and the systematic multiscale numerical analysis of their emergent dynamics. The specific tasks of interest here include the detection of tipping points, and the uncertainty quantification of rare events near them. Our illustrative examples are an event-driven, stochastic financial market model describing the mimetic behavior of traders, and a compartmental stochastic epidemic model on an Erdös-Rényi network. We contrast the pros and cons of the different types of surrogate models and the effort involved in learning them. Importantly, the proposed framework reveals that, around the tipping points, the emergent dynamics of both benchmark examples can be effectively described by a one-dimensional stochastic differential equation, thus revealing the intrinsic dimensionality of the normal form of the specific type of the tipping point. This allows a significant reduction in the computational cost of the tasks of interest.

Complex systems are typically characterized by multiscale phenomena giving rise to unexpected emergent behavior[1,2] including catastrophic shifts/major irreversible changes in the dominant mesoscopic/macroscopic spatio-temporal behavioral pattern. Such sudden major changes occur with higher probability near so-called *tipping points*[3–6], which are most often associated with bifurcation points in nonlinear dynamics terminology. The computation of the frequency/probability of occurrence of such transitions, and the detection of the corresponding tipping points that underpin them, is of critical importance in many real-world systems. Our need for understanding (and controlling) such phenomena has made Agent-Based Models (ABMs) a key modeling tool for building digital twins in domains ranging from ecology[7–10] and epidemics[11–16], to finance and economy[17–20]. Examples include the models of infectious disease agent study (MIDAS) research

network, initiated in 2004 by the US National Institutes of Health (NIH) with the mission to develop large-scale ABMs to understand infectious disease dynamics and assist policymakers to detect, and respond to flu pandemics; the Santa-Fe Artificial Stock Market; and, the Eurace ABM of the European economy[19,21]. When such detailed high-fidelity ABMs are available, systems-level analysis practice often involves performing extensive, brute-force temporal simulations to estimate the frequency distribution of abrupt transitions[17,19,20,22,23]. However, such an approach confronts the "curse of dimensionality": the computational cost rises exponentially with the number of degrees of freedom[17,20]. Such a direct simulation scenarios approach is therefore neither systematic nor computationally efficient for high-dimensional ABMs. Furthermore, it often does not provide physical insight regarding the mechanisms that drive the transitions. A systematic analysis of such mechanisms

[1]Modelling Engineering Risk and Complexity, Scuola Superiore Meridionale, Naples, Italy. [2]Department of Chemical and Biomolecular Engineering, Johns Hopkins University, Baltimore, MD, USA. [3]Department of Applied Mathematics and Statistics, Johns Hopkins University, Baltimore, MD, USA. [4]Department of Mechanical Engineering, Johns Hopkins University, Baltimore, MD, USA. [5]Dipartimento di Matematica e Applicazioni 'Renato Caccioppoli', Università degli Studi di Napoli Federico II, Naples, Italy. [6]School of Medicine's Dept. of Urology, Johns Hopkins University, Baltimore, MD, USA. ✉e-mail: constantinos.siettos@unina.it; yannisk@jhu.edu

requires two essential tasks. First comes the discovery of an appropriate low-dimensional set of collective variables (observables) that can be used to describe the evolution of the emergent dynamics[24,25]. Such coarse-scale variables may, or may not, be a priori available, depending on how much physical insight we have about the problem. For this task, various manifold/machine learning methods have been proposed, including diffusion maps (DMAPs)[26-30], ISOMAP[31,32], and local linear embedding (LLE)[33,34], but also autoencoders (AE)[35-38].

Based on this initial analysis, the second task pertains to the construction of appropriate reduced-order models (ROMs), in order to parsimoniously perform useful numerical tasks and—hopefully—obtain additional physical insight. One option is the construction of ROMs "by paper and pencil", using the tools of statistical mechanics[24,25]. However, restrictive assumptions, that are made in order to obtain explicit closures bias the estimation of the actual location of tipping points, as well as the statistics and the uncertainty quantification (UQ) of the associated catastrophic shifts[24].

Another option is the direct, data-driven identification of surrogate models in the form of ordinary, stochastic, or partial differential equations via machine learning. Such approaches include, to name a few, sparse identification of nonlinear dynamical systems (SINDy)[39], Gaussian process regression (GPR)[29,40,41], feedforward neural networks (FNNs)[29,30,38,42-46], random projection neural networks (RPNNs)[30,47], recursive neural networks (RvNN)[37], reservoir computing (RC)[48], autoencoders[37,38,49,50], as well as DeepOnet[51]. However, their approximation accuracy clearly depends very strongly on the available training data, especially around the tipping points, where the dynamics can even blow up in finite time.

If the coarse-variables are known, the Equation-free (EF) approach[1] offers an efficient alternative for learning "on demand" *local* black-box coarse-grained maps for the emergent dynamics on an embedded low-dimensional subspace; this bypasses the need to construct (global, generalizable) surrogate models. This approach can be particularly useful when conducting numerical bifurcation analysis, or designing controllers for ABMs[52]. However, even with a knowledge of good coarse-scale variables, constructing the necessary *lifting operator* (going from coarse scale descriptions to consistent fine scale ones) is far from trivial[52,53].

Here, based on our previous efforts on the construction of latent spaces[52,54,55] and ROM surrogates via machine learning (ML)[29,30,34,45,46] from microscopic detailed spatio-temporal simulations, we present an integrated ML framework for the construction of two types of surrogate models: global as well as local. In particular, we learn (a) mesoscopic Integro Partial Differential Equations (IPDEs), and (b)—guided by the EF framework[1,56]—local embedded low-dimensional mean-field Stochastic Differential Equations (SDEs), for the detection of tipping points and the construction of the probability distribution of the catastrophic transitions that occur in their neighborhood.

Our main methodological point is that, given a macroscopic task, it is the task itself that determines the type of surrogate model required to perform it. Here, the tasks are the identification of tipping points as well as the uncertainty quantification of escape times in their neighborhood. For such tasks, a first option—common in practice—is the construction of a data-driven, ML-identified IPDEs for macroscopic fields, such as the agent density. Such equations provide some physical insight for the emergent dynamics, yet this insight does not come without its problems: for example, collecting the training data and designing the sampling process in the relatively high dimensional parameter and state space is not an easy task, especially in the unstable regimes where the dynamics may blow up in finite time. The second option, assuming an approximate knowledge of the tipping point location, is to identify a less detailed, mean-field-level, effective SDE. This offers the capability of more easily estimating escape time statistics through either: (a) brute-force bursts of SDE simulations, or

through (b) numerically solving a boundary-value problem given the identified low-dimensional drift and diffusivity functions. For such approach, a challenging issue is the discovery of a convenient/interpretable low-dimensional latent-space.

Thus, in the spirit of the Wykehamist "manners makyth man", and of Rutherford Aris' "manners makyth modellers"[57], in our case, we argue for a "tasks makyth models" consideration in selecting the right approach for ML-assisted model selection[57].

Our illustrative case studies are (i) an event-driven stochastic agent-based model describing the interactions, under mimesis of traders in a simple financial market[58]. In other words, the traders in this ABM tend to imitate the behavior of other traders, because of social conformity or subtle psychological pressure to align their behavior with that of other agents (their peers). This model exhibits a tipping point, marking the onset of a financial "bubble"[23,56]; (ii) a stochastic ABM of a host-host interaction epidemic evolving on an Erdös-Rényi social network[59]. This ABM is characterized by a tipping point marking the onset of outbreaks and regions of hysteresis, where transitions between "endemic disease" and "global infection" states can occur.

The proposed ML framework reveals that the emergent dynamics of both ABMs around the tipping points can be effectively described on a one-dimensional manifold. In other words, it discovers the intrinsic dimensionality of the normal form of the specific type of tipping point, which for both problems is a saddle-node bifurcation. This allows for a significant reduction of the computational cost required for numerical analysis and simulations.

## Results
### Case study 1: Tipping points in a financial market with mimesis
ABMs enable the creation of digital twins for financial markets, thus offering a valuable tool in our arsenal for explaining out-of-equilibrium phenomena such as "bubbles" and crashes[17] that emerge mainly due to positive feedback mechanisms of imitation and herding of investors that lead to an escalating increase of the demand[60] (see for example the Santa Fe artificial stock market[19,61], and the EURACE ABM for modeling the European economy[21]). While the practical application of ABMs for providing predictions about real-world financial instabilities remains an ongoing area of research, they can be used to shed light on the mechanisms that lead to such crises[17,60]. Towards to this aim, our first illustrative example is an event-driven agent-based model approximating the dynamics of a simple financial market with mimesis proposed by Omurtag and Sirovich[58]. The ABM describes the interactions of a large population of, say $N$, financial traders. Each agent is described by a real-valued state variable $X_i(t) \in (-1, 1)$ associated to their tendency to buy (positive values) or sell (negative values) stocks in the financial market according to constantly updated financial news, as well as to their interactions with the other traders[58]. The $i$-th agent acts, i.e., buys or sells, only when its state $X_i$ crosses one of the decision boundaries/thresholds $X = \pm 1$. As soon as an agent $i$ buys or sells, the agent's state is forthwith reset to zero.

In the absence of any incoming good news $I_i^+$ or bad news $I_i^-$, the preference state exponentially decays to zero with a constant rate $\gamma$. Thus, each agent is governed by the following SDE:

$$dX_i(t) = -\gamma X_i(t)dt + dI_i^+(t) + dI_i^-(t), |X_i| < 1. \qquad (1)$$

The effect of information arrivals $I_i^\pm(t)$ is represented by a series of instantaneous positive/negative "discrete jumps" of size $\epsilon^\pm$, arriving randomly at Poisson distributed times $t_{k^+}, k^+ = 1, 2, \ldots$ and $t_{k^-}, k^- = 1, 2, \ldots$, with average rates of arrival $\nu^+(t)$ and $\nu^-(t)$, respectively. Furthermore, the dynamics of each agent are driven by arrivals of two types of information: *exogenous* (*ex*) (e.g., publicly available financial news), as well as an *endogenous* (*en*) stream of information

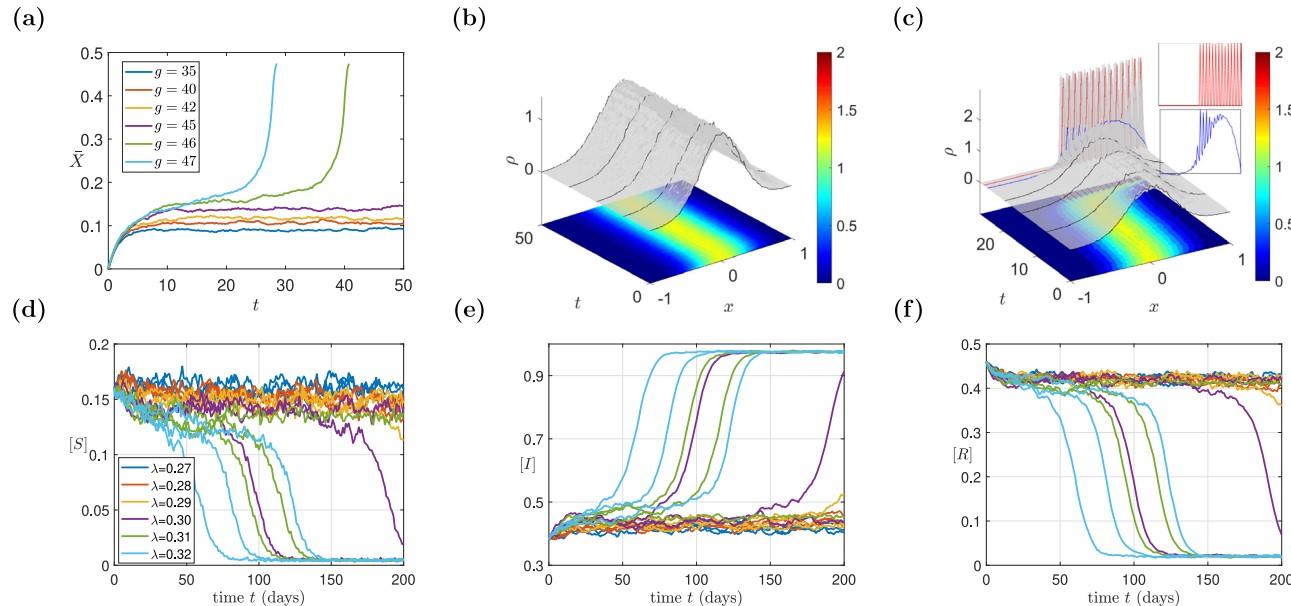

**Fig. 1 | Stochastic Agent-based model simulations of the two case studies. a–c** traders in a simple financial market. **a** Trajectories for different values of the parameter $g$. **b** Probability density function (pdf) evolution for $g = 45$; **c** pdf evolution for $g = 47$ (past the tipping point); Insets show the blow up of the pdf; the blue curve depicts the pdf just a few time steps before the explosion and the red curve depicts the pdf at the financial "bubble". **d–f** Stochastic simulations of the epidemic ABM. Trajectories of the densities [S] in (**d**), [I] in (**e**), and [R] in (**f**), for different values of the parameter $\lambda$.

arising from the social connections of the agents, so that

$$\nu^{\pm} = \nu^{\pm}_{ex} + \nu^{\pm}_{en}. \tag{2}$$

A tunable parameter $g$ embodies *the strength of mimesis*: the extent to which arriving information affects the willingness or apprehension of the agent to buy or sell. For this model, the term $\nu^{\pm}_{en}$ is set to be the same for all agents and is influenced by the perceived overall buying $R^{+}(t)$ and selling $R^{-}(t)$ rates:

$$\nu^{\pm}(t) = \nu^{\pm}_{ex} + gR^{\pm}(t), \tag{3}$$

where $R^{\pm}(t)$ are defined as the fraction of agents buying or selling per unit of time $\Delta t$:

$$R^{\pm}(t) = \frac{\text{number of agents buying/selling}}{\Delta t \cdot \text{total number of agents}} = \\ = \frac{1}{N\Delta t}\int_{t}^{t+\Delta t}\delta(s - T_i^{\pm})ds, \tag{4}$$

where $T_i^{\pm}$ are the instants at which the $i$-th agent crosses the decision boundary $\pm 1$.

In Fig. 1a, we depict the mean preference state for $N = 50,000$ agents for values of the mimesis strength $g = 35, 40, 42, 45, 46, 47$. In Fig. 1b, c, we depict representative trajectories of the time evolution of the agent probability density distribution (pdf) for $g = 45$ and $g = 47$, respectively. We see that the simulations exhibit a tipping point that arises at a parameter value $g \approx 45.5$. At the neighborhood of this tipping point, due to the inherent stochasticity of the mimetic trading process, emanate "financial bubbles", where all agents hurry to buy assets (see Fig. 1a). The ABM model also predicts financial crashes in regimes of the phase-space where the mean value of the mesoscopic density field is negative, and the agents rush to sell (for more details see ref. 56).

A concise analytical mesoscopic description of the population dynamics was derived by Omurtag and Sirovich in ref. 58. The model, reported here, is a Fokker-Planck-type (FP) IPDE for the agent pdf $\rho(x, t)$, given by:

$$\frac{\partial \rho(t,x)}{\partial t} = \frac{1}{2}\sigma^2(t)\frac{\partial^2 \rho(t,x)}{\partial x^2} + \frac{\partial(\mu(t,x)\rho(t,x))}{\partial x} + \\ + (J^{+} + J^{-})\delta(x). \tag{5}$$

where $\mu$ and $\sigma$ are drift and diffusivity time-dependent parameters, respectively, $\delta$ is the Dirac delta and $J^{\pm}$ are integral operators accounting for the agents crossing the decision boundaries.

Further details about the derivation of the FP equation (5) are presented briefly in the section A of the Supplementary Information (SI).

## ML mesoscopic IPDE surrogate for the financial ABM

In ref. 56, we showed that the analytical ROM IPDE in Eq. (5) nontrivially underestimates the location of the tipping point with respect to the parameter $g$, defined in Eq. (3). Here we show how one can achieve a better approximation through data-driven black-box surrogates. Based on data generated as in section E1 of the Supplementary Information (SI), for learning the right-hand-side operator of the IPDE, we have considered the relevant features that we found with Automatic Relevance Determination (see the section "Methods" and section E2 of the Supplementary Information). We used the following (black-box) mesoscopic model for the dynamic evolution of the density $\rho$:

$$\frac{\partial \rho(x,t)}{\partial t} = F\left(x, \rho(x,t), I^{+}, I^{-}, \frac{\partial \rho(x,t)}{\partial x}, \frac{\partial^2 \rho(x,t)}{\partial x^2}; g\right) \tag{6}$$

where $I^{+}, I^{-}$ are integrals in a small neighborhood of the boundaries (see section A of the Supplementary information (SI)). Here, for learning the RHS of the black-box IPDE (6), we implemented two different structures, namely (a) a feedforward neural network (FNN)[62–64]; and (b) a Random Projection Neural Network[47,65–68] in the form of Random Fourier Feature network (RFF)[69]. The two network structures and their training protocols are described in more detail in the sections D2 and D3 of the Supplementary information (SI). The two alternative machine learning schemes, on the test set, obtain similar performance in terms of accuracy. In terms of the mean absolute error (MAE) the FNN got 1.10E−04 and the RFF got 1.09E−04. For mean squared error (MSE), the

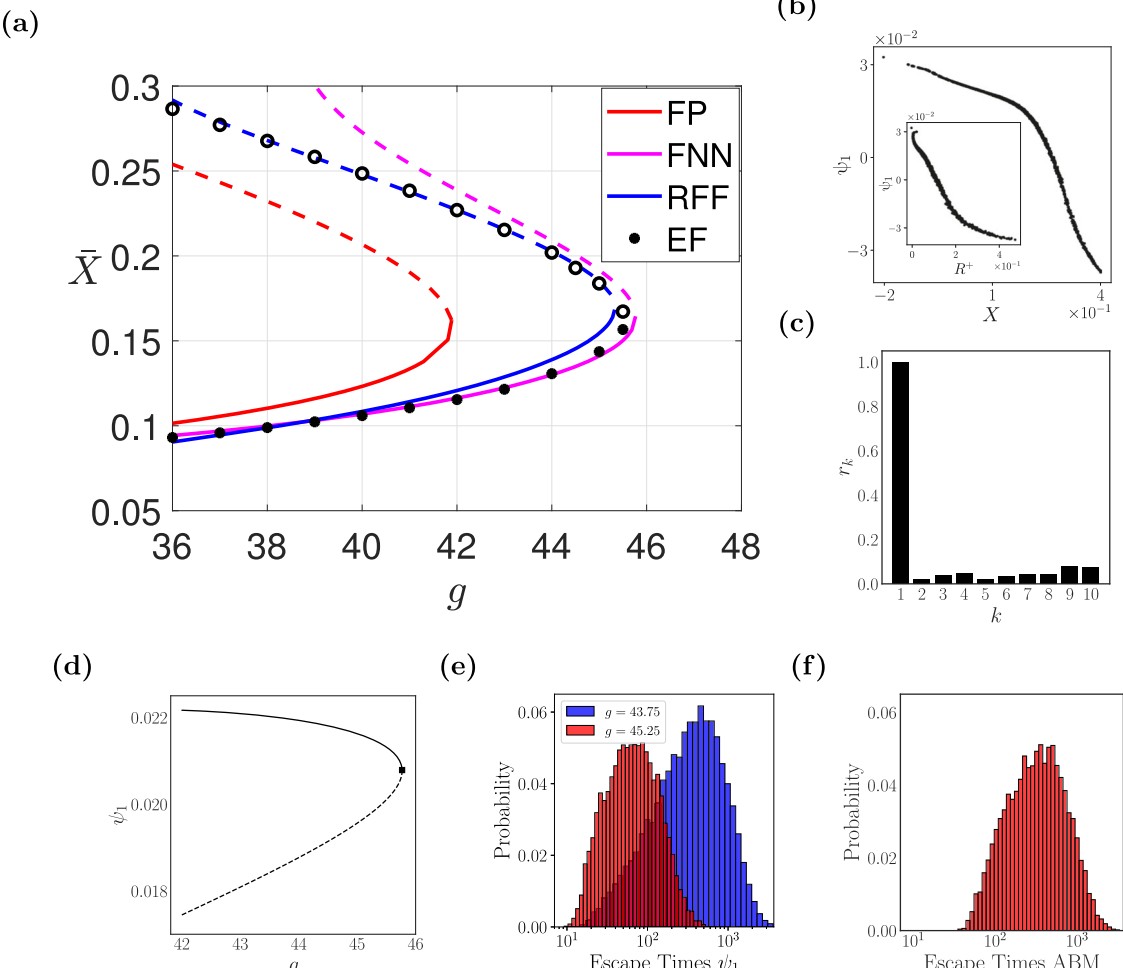

**Fig. 2 | Numerical results for the financial ABM. a** Reconstructed bifurcation diagram w.r.t. $g$ obtained with the mesoscopic IPDE surrogate FNN and RFF models; the one computed from the analytical Fokker-Planck (FP) IPDE, see Eq. (5), and the one constructed with the Equation-free (EF) approach are also given[52,56]. Dashed lines (open circles) represent the unstable branches. **b**, **c** ABM-based-simulation- and Diffusion Maps- (DMAPs) driven observables. **b** The first DMAPs coordinate $\psi_1$ is plotted against the mean preference state $\bar{X}$. In the inset, the buying rate $R^+$ plotted against the mean preference state ($\bar{X}$). **c** The estimated residual $r_k$ based on the local linear regression algorithm[70]. **d** The effective bifurcation diagram based on the drift component of the identified mean-field macroscpic SDE in $\psi_1$. **e**, **f** Histograms of escape times obtained with simulations of 10,000 stochastic trajectories for (**e**) the SDE model for $g = 45.25$ (blue histogram) and $g = 43.75$ (red histogram) (**f**) the full ABM at $g = 45.25$.

FNN got 2.60E−08 and the RFF got 2.53E−08. For the Regression Pearson correlation $R$, FNN got 0.9866 and RFF 0.9873. The main notable difference between the two schemes is in the computational time needed to perform the training, since remarkably, the training of the RFF, which required 26.06 (s), turned out to be at least 50 times faster than the one required for the deep-learning scheme, which required 1488.33 (s).

## Bifurcation analysis of the mesoscopic IPDE for the financial ABM

To locate the tipping point, we have performed bifurcation analysis, using both ML-identified IPDE surrogates, as discussed in section "Methods". Furthermore, we compared the derived bifurcation diagram(s) and tipping point(s) with what was obtained in refs. 52,56 using the EF approach (see in the section C of the Supplementary Information (SI) for a very brief description of the EF approach). As shown in Fig. 2a, the two ML schemes approximate visually accurately the location of the tipping point in parameter space. However, the FNN scheme fails to trace accurately the actual coarse-scale unstable branch, near which simulations blow up extremely fast. More precisely, the analytical FP predicts the tipping point at $g^* = 41.90$ with

corresponding steady-state $\bar{X}^* = 0.1607$ and the EF at $g^* = 45.60$ and $\bar{X}^* = 0.1627$; our FNN predictions are at $g^* = 45.77$ and $\bar{X}^* = 0.1644$, the RFF ones at $g^* = 45.34$ and $\bar{X}^* = 0.1684$.

## Macroscopic physical observables and latent data-driven observables via DMAPs

An immediate physically meaningful candidate observable is the first moment $\bar{X}$ of the agent distribution function (as also shown in ref. 23).

As simulations of the ABM show (see the inset in Fig. 2b), the mean preference state $\bar{X}$, is one-to-one with another physically meaningful observable, the buying rate $R^+$. We also used the DMAPs algorithm, to discover *data-driven* macroscopic observables. In our case, DMAPs applied to collected data (see section F1 of the Supplementary Information (SI) for a detailed description of how the data were collected), discovers a 1D latent variable $\psi_1$ that is itself one-to-one with $\bar{X}$, see Fig. 2b. The local-linear regression algorithm proposed in[70] was applied to make sure that all the higher eigenvectors can be expressed as local-linear combinations of $\psi_1$ and thus they do not span independent directions. Figure 2c illustrates that the normalized leave-one-out error, denoted as $r_k$, is small for $\psi_2, ..., \psi_{10}$ suggesting they are all dependent/harmonics of $\psi_1$.

**Table 1 | Escape time computations for the financial ABM**

| Models | SDE at $g = 45.25$ | SDE at $g = 43.75$ | ABM |
|---|---|---|---|
| Mean Escape Time | 84.07 | 480.92 | 434.00 |
| Escape Time Standard deviation | 68.91 | 454.83 | 363.64 |

Means and Standard deviations as computed with temporal simulations from the SDE trained on the DMAPs variable $\psi_1$ for $g = 45.25$, $g = 43.75$ and the ABM at $g = 45.25$, respectively.

Therefore, any of the three macroscopic observables (two physical and one data-driven) can be interchangeably used to study the collective behavior of the model.

### Learning the mean-field SDE and performing bifurcation analysis for the financial ABM

Here, for our illustrations, we learned parameter-dependent SDEs for all of the three coarse variables we mentioned, namely the physically meaningful variables $\bar{X}$, $R^+$, and the DMAPs coordinate $\psi_1$. In the main text, we report the results for the identified SDEs in terms of the DMAPs coordinate $\psi_1$ and in section F3 of Supplementary Information (SI) we report the results for the identified SDEs with respect to the $\bar{X}$ and $R^+$ (see Supplementary Fig. 1) .

Given this trained macroscopic SDE surrogate, the drift term (deterministic component) of the identified dynamics was used to construct the bifurcation diagram with AUTO[71] (see Fig. 2d). A saddle-node bifurcation was identified for $g^* = 45.77$ where $\psi_1^* = 0.021$. The estimated critical parameter value from the SDE is in agreement with our previous work ($g \approx 45.60$[56]). Details pertaining to the neural networks' architectures used to identify the SDE are provided in section F5 of Supplementary Information (SI).

### Rare-event analysis/UQ of catastrophic shifts (financial "bubbles") via the identified mean-field SDE

Given the identified steady states at a fixed value of $g$ we performed escape time computations. For $g = 45.25$, we estimated the average escape time needed for a trajectory initiated at the stable steady state to reach $\bar{X} = 0.3$, i.e., sufficiently above the unstable branch. As shown in Fig. 2b, $\psi_1$ and $R^+$ are effectively one-to-one with $\bar{X}$, and we can easily find the corresponding critical values for $\psi_1 = -0.01$ (flipped) and $R^+ = 0.16$. We now report a comparison between the escape times of an SDE identified based on the DMAPs coordinate $\psi_1$ and those of the full ABM. In section F3 of Supplementary Information (SI) we also report the escape times of the SDE for $\bar{X}$ and $R^+$ observables. To estimate these escape times we sampled a large number (10,000 in our case) of trajectories. In section F4 of Supplementary Information (SI) we also include the escape time computation by using the closed-form formula for the 1D case. The computation there was performed numerically by using quadrature and the milestoning approach[22].

In Fig. 2e, the histograms of the escape times for the identified SDE trained on $\psi_1$ for $g = 45.25$ and $g = 43.75$ are shown. In Fig. 2f, we also illustrate the empirical histogram of escape times of the full ABM for $g = 45.25$. The estimated values for the mean and standard deviation, as computed with temporal simulations from the SDE trained on the DMAPs variable $\psi_1$ for $g = 45.25$, $g = 43.75$ and the full ABM at $g = 45.25$ are here reported in Table 1.

As shown, the mean escape time of the full ABM is a factor of five larger than that estimated by the simplified SDE model in $\psi_1$ for $g = 45.25$ (still within an order of magnitude!). The SDE model for $g = 43.75$ gives an escape time comparable to the one of the ABM for $g = 45.25$. Given that the escape times change exponentially with respect to the parameter distance from the actual tipping point, a small error in the identified tipping point easily leads to large (exponential) discrepancies in the estimated escape times.

### Computational cost for the financial ABM

We compared the computational cost required to estimate escape times with many stochastic temporal simulations, through the full ABM and the identified mean-field SDE. To fairly compare the computational costs, we computed the escape times with the ABM for $g = 45.25$, and that of the SDE for $g = 43.75$, since the two distributions of the escape times are more comparable. The estimation in both cases was conducted on *Rockfish* (a community-shared cluster at Johns Hopkins University) by using a single core with 4GB RAM. For the 10,000 sampled stochastic trajectories, the total computational for the identified coarse SDE in $\psi_1$ was 33.56 min and the average time per trajectory, $3.36 \times 10^{-3}$ min. The mean time per function evaluation was approximated as the ratio of mean time per trajectory over mean number of iterations.

For the ABM, the total computational time needed was 18.56 days and the mean time per trajectory was 2.67 min. Therefore, the total computational time for computing the escape time with the SDE model in $\psi_1$ was around 800 times faster than the ABM. This highlights the computational benefits of using the reduced surrogate models in lieu of the full ABM for escape time computations.

### Case study 2: Tipping points in a compartmental epidemic model on a complex network

In our second illustrative example, a compartmental epidemic ABM on a social network[59], individuals are characterized by three discrete states: Susceptible ($S$), Infected ($I$) and Recovered ($R$). The model is implemented through a "caricature of a social network" approximated by an Erdos-Rényi network with $N = 10,000$ nodes. The probability of having a connection between two nodes picked randomly is $p = 0.0008$ (as proposed in[59]). The evolution rules are the following:

- Rule 1 ($S \to I$): Susceptible individuals may become infected upon contact with infected individuals, with probability $P_{S \to I} = \lambda$. This tunable parameter is "tracked" for studying the outcomes of abrupt changes in the macroscopic behavior.
- Rule 2 ($I \to R$): The transition between $I$ and $R$ happens with a probability $P_{I \to R} = \mu([I])$. The probability of recovery depends, at each time step, on the overall density of infected individuals $[I]$, according to the function[59]:

$$\mu([I]) = 0.3 \left(1 - \frac{1}{1 + \exp(-9([I] - 0.5))}\right). \quad (7)$$

Such a nonlinear function for the probability of recovery has also been used in other works to express the heterogeneity in the "environment" around each individual (see also the discussion in ref. 59).

- Rule 3 ($I \to R$): A recovered individual ($R$) loses its immunity and becomes susceptible ($S$) with a fixed probability $P_{R \to S} = \epsilon = \frac{1}{5}$. This condition expresses the case of temporal immunity.

For further details about the construction of the Erdös-Rényi network, see section B of the Supplementary Information (SI).

The above rules establish a complex stochastic microscopic model that change the state of each individual over time. In order to describe the model at a macroscopic (emergent) level, let us represent the overall density of susceptible, infected, and recovered individuals as $[S]$, $[I]$, and $[R]$, respectively. In Fig. 1d–f, we depict the stochastic trajectories of the overall densities $[S]$, $[I]$, $[R]$ of the $N = 10,000$ agents for values of the rate of infection $\lambda = 0.27, 0.28, 0.29, 0.30, 0.31, 0.32$. For such macroscopic observables, a simple analytical closure can be found assuming that a *uniform and homogeneous network* is a good approximation. This means assuming that (a) the degree of each node

practically coincides with the mean degree of the network, denoted as $z = \mathbb{E}(k)$; (b) that the probabilities of two connected nodes being in a susceptible and infected state, respectively, are independent of each other. The resulting mean field model reads[59]:

$$
\begin{aligned}
\frac{d[S]}{dt} &= -\lambda z[S][I] + \epsilon[R], \\
\frac{d[I]}{dt} &= \lambda z[S][I] - \mu([I])[I], \\
&[S] + [I] + [R] = 1.
\end{aligned}
\tag{8}
$$

For other higher-order analytical macroscopic pairwise closures, such as the Bethe Ansatz, the Kirkwood approximation and Ursell expansion, the interested reader can consult[59].

## ML macroscopic mean-field surrogates for the epidemic ABM

It is well known that for dynamics evolving on complex networks, a closed-form, analytically derived mean-field approximation in Eq. (8) is usually not accurate[59]. Here, we show how one can achieve a better approximation through the construction of effective mean field-level ML surrogate models. We will start with the identification, from data, of a mean field-level effective SIR model. Then, following the proposed approach, we identify—again from data—an effective one-dimensional SDE to model the stochastic dynamics close to the tipping point and quantify the probability of occurrence of an outbreak where all the population becomes infected.

Given the high-fidelity data collected from the epidemic ABM, as described in section G1 of the Supplementary Information (SI), to learn the ML mean field SIR surrogate we used two coupled feedforward neural networks (FNN), labeled $F_S$ and $F_I$, each with two hidden layers with 10 neurons for each layer, for learning a black-box evolution for the effective dynamics of the two macroscopic densities $[S]$ and $[I]$, that reads:

$$
\begin{aligned}
\frac{d[S]}{dt} &= F_S([S],[I],[R];\lambda), \\
\frac{d[I]}{dt} &= F_I([S],[I],[R];\lambda),
\end{aligned}
\tag{9}
$$

with the constraint $[S] + [I] + [R] = 1$. We remind the reader that the parameter $\lambda$, representing the probability of a susceptible individual to get infected, is tracked for bifurcation analysis purposes. The training process (see sections D2 and D3 of Supplementary Information (SI)) results to a MAE of $7.27e-04$ and a MSE of $1.27e-06$ on the test set. The regression error for the two networks was $R(F_S) = 0.9996$ and $R(F_I) = 0.9992$.

## Bifurcation analysis of the ML mean-field SIR surrogates

To locate the tipping point, we have performed bifurcation analysis using the ML mean-field SIR surrogates. For our illustration, we also compare the derived bifurcation diagram(s) and tipping point(s) with those obtained in[59] using the EF approach and the analytically derived mean-field SIR model given by Eq. (8). As shown in Fig. 3a, the ML SIRS surrogate approximates adequately the location of the tipping point in parameter space, as well as the entire bifurcation diagram as constructed with the EF approach. Additionally, it outperforms the statistical-mechanics-derived mean-field approximation given by Eq. (8). More precisely, the statistical-mechanics-derived mean-field SIRS model predicts the tipping point at $\lambda^* \approx 0.166$ with corresponding steady-state $([S]^*, [I]^*) = (0.138, 0.449)$ and the EF at $\lambda^* = 0.289$ with corresponding steady-states $([S]^*, [I]^*) = (0.138, 0.451)$; our ML mean-field SIR surrogate predictions are at $\lambda^* = 0.304$ with corresponding steady-states $([S]^*, [I]^*) = (0.135, 0.456)$.

## Macroscopic physical observables and data-driven observables via DMAPs for the epidemic ABM

Two immediate physically meaningful candidate observables are the densities $[S]$, $[I]$. However, close to the saddle node bifurcation the system is effectively one-dimensional and $[S]$ and $[I]$ are effectively one-to-one with each other in the long-term dynamics, eventually taking place on the slow eigenvector of the stable steady state.

We also demonstrate this via the DMAPs algorithm. In our case, DMAPs applied to the collected data (see section G2 of the Supplementary Information (SI) for more details), discovers a one-dimensional latent variable parameterized by $\phi_1$ that is also one-to-one with $[I]$ (see Fig. 3b). The local-linear regression algorithm proposed in ref. 70 was also applied to confirm that the remaining eigenvectors are harmonics of $\phi_1$ and thus they do not span independent directions, see Fig. 3c. This confirms that the emergent ABM dynamics close to the tipping point lie on one-dimensional manifold.

## Learning the effective mean-field-level SDE and performing bifurcation analysis for the epidemic ABM

Here, for our illustration, we learned a one-dimensional parameter-dependent SDE for the DMAPs coordinate $\phi_1$.

Given this trained macroscopic SDE surrogate, the drift term (deterministic component) of the identified dynamics was used to construct the bifurcation diagram with AUTO (see Fig. 3d). A saddle-node bifurcation was identified for $\lambda^* = 0.294$ where $\phi_1^* = 0.006$. The estimated critical parameter value from the SDE is in agreement with our previous work ($\lambda \approx 0.289$[59]).

## Rare-event analysis/UQ of catastrophic shifts via the identified SDE

Given the identified steady states at a fixed value of $\lambda$ we performed escape time computations. For $\lambda = 0.29$, we estimated the average escape time needed for a trajectory initiated at the stable steady state to reach $[I] = 0.662$, a value sufficiently above the unstable branch. The corresponding value for the DMAPs coordinate was $\phi_1 = -0.007$. We now report a comparison between the escape times of the SDE identified based on the DMAPs coordinate $\phi_1$ and those of the epidemic ABM. For this model we also sampled 10,000 stochastic trajectories.

In Fig. 3e, the histograms of the escape times for the SDE trained on $\phi_1$ are shown for $\lambda = 0.29$ and $\lambda = 0.285$. For the full epidemic ABM in Fig. 3f, we depict the histogram of escape times for $\lambda = 0.29$.

The estimated values for the mean and standard deviation, as computed with temporal simulations from the SDE trained on the Diffusion Map variable $\phi_1$ for $\lambda = 0.29$, are reported in Table 2.

As shown, the mean escape time of the full ABM is four times larger than that estimated by the simplified SDE model for $\lambda = 0.29$ (still within an order of magnitude). The SDE model for $\lambda = 0.285$ gives an escape time comparable to the one of the ABM for $\lambda = 0.29$.

As we mentioned earlier, the escape times change exponentially depending on the parameter value. Therefore, a small error in the estimated location of the tipping point can lead to large discrepancies in the estimated escape times.

## Computational cost for the epidemic ABM

We compared the computational time required to estimate escape times with the full ABM and the identified mean-field SDE. To compare the computational times, we computed the escape times with the ABM for $\lambda = 0.29$, and that of the SDE for $\lambda = 0.285$, since the mean escape times there are more similar. The estimation of the computational cost for both models (SDE and ABM) was conducted in Matlab. For the 10,000 stochastic trajectories, the total computational time for the identified coarse SDE was -1 min and the one for the full epidemic ABM was -16 h.

## Discussion

Performing uncertainty quantification of catastrophic shifts, designing control policies for them[52,56] and thus eventually preventing them, is one of the biggest challenges of our times. Climate change and extinction of ecosystems, outbreak of pandemics, economical crises, all can be attributed to both systematic changes and stochastic

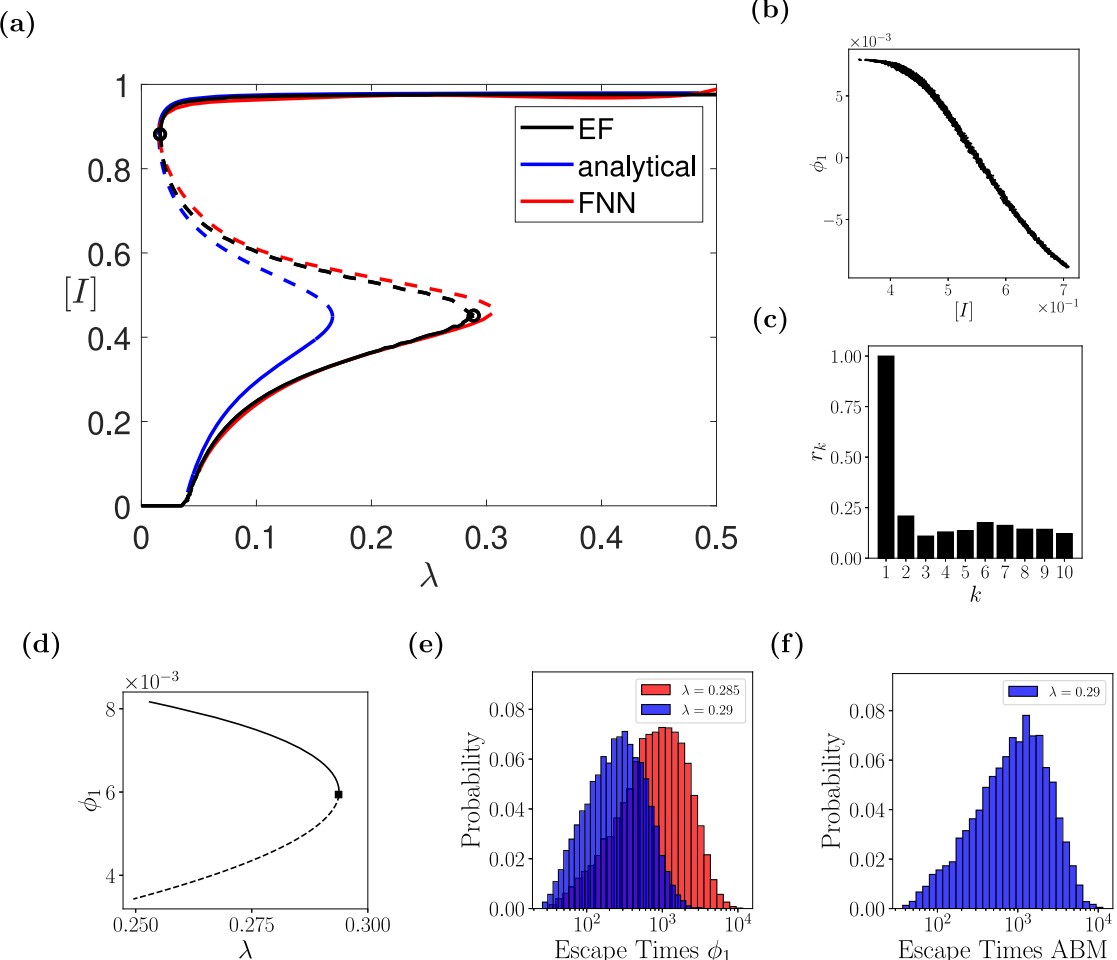

**Fig. 3 | Numerical results for the epidemic ABM. a** Reconstructed bifurcation diagram w.r.t. the probability of infection, $\lambda$, with the ML mean-field surrogate model; the one computed from the analytical mean-field Eq. (8), and the one constructed with the Equation-free (EF) approach are also given[59]. Dashed lines represent the unstable branches. **b, c** ABM-based simulations- and Diffusion Maps- (DMAPs) driven observables. **b** The density of infected [$I$] vs. the first DMAPs coordinate $\phi_1$. **c** The estimated residual $r_k$ based on the local linear regression algorithm[70]. **d** The effective bifurcation diagram based on the drift component of the identified mean-field macrosocpic SDE based on $\phi_1$. **e, f** Histograms of escape times obtained with simulations of 10,000 stochastic trajectories using: **e** the constructed SDE for $\lambda = 0.29$ (blue histogram), $\lambda = 0.285$ (red histogram), and **f** the full ABM at $\lambda = 0.29$.

perturbations that, close to tipping points, can drive the system abruptly towards another regime that might be catastrophic. Such tipping points are most often associated with underlying bifurcations. Hence, the systematic identification of the mechanisms—types of bifurcations—that govern such shifts, and the quantification of their occurrence probability, is of utmost importance. Towards this aim, as real experiments in the large scale can be difficult, or impossible to perform (not least due to ethical reasons) mathematical models and especially high-fidelity large-scale agent-based models are a powerful tool in our arsenal to build informative "digital twins" (see also the discussion in refs. 3,4). However, due to the "curse of dimensionality" that pertains to the dynamics of such large-scale "digital twins", the above task remains computationally demanding and challenging.

**Table 2 | Escape time computations for the epidemic ABM**

| Models | SDE at $\lambda = 0.29$ | SDE at $\lambda = 0.285$ | ABM |
|---|---|---|---|
| Mean Escape Time | 348.24 | 1164.65 | 1308.65 |
| Escape Time Standard deviation | 310.16 | 1117.56 | 1247.70 |

Means and standard deviations as computed with temporal simulations from the SDE trained on the Diffusion Map variable $\phi_1$ for $\lambda = 0.29$, $\lambda = 0.285$ and the ABM at $\lambda = 0.29$, respectively.

Here, we proposed a machine-learning-based framework to infer tipping points in the emergent dynamics of large-scale agent-based simulators. In particular, we proposed and critically discussed the construction of mesoscopic and coarser/mean-field-level ML surrogates from high-fidelity spatio-temporal data for: (a) the location of bifurcation points and their type, and (b) the quantification of the probability distribution of the occurrence of the catastrophic shifts. As our illustration, we used two large-scale ABMS: (1) an event-driven stochastic agent-based model describing the mimetic behavior of traders in a simple financial market, and (2) an epidemic ABM on a complex social network. In both ABMs tipping points arise, which give rise to financial bubbles and epidemic outbreaks, respectively. While analytical surrogates may provide some physical insight for the emergent dynamics, they introduce biases in the accurate numerical bifurcation analysis, and thus also in the accurate detection of tipping points, especially when dealing with IPDEs. On the other hand, discovering sets of variables that span the low-dimensional latent space on which the emergent dynamics emerge—via manifold learning—and then learning surrogate mean-field-level SDEs in these variables, offers an attractive and computationally "cheap" alternative. Such an approach for the construction of tipping point-targeted ROMs can provide an accurate approximation of the tipping point. However, this

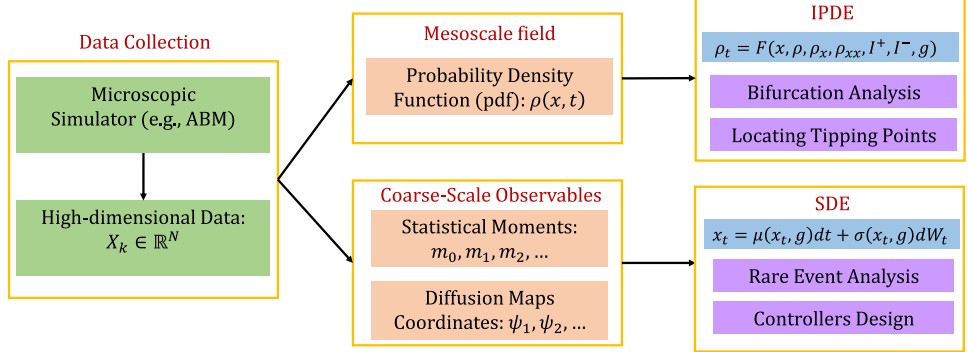

**Fig. 4 | Schematic of the machine learning-based approach for the multiscale modeling and analysis of tipping points.** At the first step, and depending on the scale of interest, we discover via Diffusion Maps latent spaces using, mesoscopic fields (probability density functions (pdf) and corresponding spatial derivatives) with the aid of Automatic Relevance Determination (ARD); or macroscopic mean-field quantities, such as statistical moments of the probability density function. At the second step, on the constructed latent spaces, we solve the inverse problem of identifying the evolutionary laws, as IPDEs for the mesoscopic field scale, or mean-field SDEs for the macroscopic scale. Finally, at the third step, based on the constructed surrogate models, we perform system level analysis, such as numerical integration at a lower computational cost, numerical bifurcation analysis for the detection and characterization of tipping points, and rare event analysis (uncertainty quantification) for the catastrophic transitions occurring in the neighborhood of the tipping points.

physics-agnostic approach, may result in macroscopic observables that are not, at least not explicitly, physically interpretable.

Clearly, different modeling tasks are best served by different coarse-scale surrogates. This underpins the importance of selecting the right ML modeling approach for different system-level tasks, being conscious of the pros and cons of the different scale surrogates.

Further extensions of our framework may include the learning of a more general class of SDEs (e.g., based on Lévy process)[72]; and possibly moving towards learning effective SPDEs or even fractional evolution operators that could lead to more informative surrogate models[25].

There is a point in noting that our work here does not focus on constructing (or validating) early warning systems based on real-world data. It is, however, conceivable that some of our data-driven, DMAPs identified collective variables may serve as candidate coordinates for such early-warning systems[73]. Our main target has been to show how one can systematically construct reduced-order models via machine learning for understanding, analyzing the mechanisms governing the emergence of tipping points, and quantifying the probability of occurrence of rare events around them, from detailed ABM simulations—a problem that can suffer from the curse of dimensionality.

## Methods

Given high dimensional spatio-temporal trajectory data acquired through ABM simulations, the main steps of the framework are summarized as follows (see also Fig. 4, for a schematic):

a. Discover low-dimensional latent spaces, on which the emergent dynamics can be described at the mesoscopic or the macroscopic scale.
b. Identify, via machine-learning, black-box mesoscopic IPDEs, ODEs, or (after further dimensional reduction), macroscopic mean-field SDEs.
c. Locate tipping points by exploiting numerical bifurcation analysis of the different surrogate models.
d. Use the identified (NN-based) surrogate mean-field SDEs to perform rare-event analysis (uncertainty quantification) for the catastrophic transitions. This is done here in two ways: (i) performing repeated brute-force simulations around the tipping points, (ii) for this effectively 1D problem, using explicit statistical mechanical (Feynman-Kac) formulas for escape time distributions.

In what follows, we present the elements of the methodology. For further details about the methodology and implementation, see in the Supplementary information (SI).

## Discovering low-dimensional latent spaces

The computational modeling of complex systems featuring a multitude of interacting agents poses a significant challenge due to the enormous number of potential states that such systems can have. Thus, a fundamental step, for the development of ROMs that are capable of effectively capturing the collective behavior of ensembles of agents is the discovery of an embedded, in the high-dimensional space, low-dimensional manifold and an appropriate set of variables that can usefully parametrize it.

Let's denote, by $\mathbf{X}_k \in \mathbb{R}^D$, $k = 1, 2, \ldots$ the high-dimensional state of the ABM at time $t$. The goal is then to project/map the high-dimensional data onto lower-dimensional latent manifolds $\mathcal{M} \subset \mathbb{R}^D$, that can be defined by a set of coarse-scale variables. The hypothesis of the existence of this manifold is related to the existence of useful ROMs and vice versa.

Here, to discover such a set of coarse-grained coordinates for the latent space, we used DMAPs[26,28,54] (see section D1 of the Supplementary Information (SI) for a brief description of the DMAPs algorithm).

For both ABMs, we have some a priori physical insight for the mesoscopic description. For the financial ABM, one can for example use the probability density function (pdf) $\rho(X)dx = \mathbb{P}(X(t) \in [x, x + dx])$ across the possible states $X_k$ in space. Thus, the continuum pdf constitutes a spatially dependent *mesoscopic* field that can be modeled by a FP IPDE as explained above. For the epidemic ABM, there is a physical insight on the macroscopic mean-field description, which is the well-known mean-field SIRS model. Multiscale macroscopic descriptions can also be constructed including higher-order closures[59]. Alternatively, one can also collect "enough" statistical moments of the underlying distribution such as the expected value, variance, skewness, kurtosis, etc. Nevertheless, the collected statistics may not automatically provide insight into their relevance in the effective dynamics and a further feature selection/sensitivity analysis may be needed.

Focusing on a reduced set of coarse-scale variable is particularly relevant when there exists a significant separation of time scales in the system's dynamics. By selecting only a few dominant statistics, one can effectively summarize the behavior of the system at a coarser level.

The choice of the scale and details of coarse-grained description, leads to different modeling approaches. For example, focusing at the *mesoscale* for the population density dynamics, we aim at constructing a FP-level IPDE for the financial ABM, and a mean-field SIR surrogate for the epidemic ABM. At an even coarser scale, e.g., for the first moment of the distribution, and taking into account the underlying

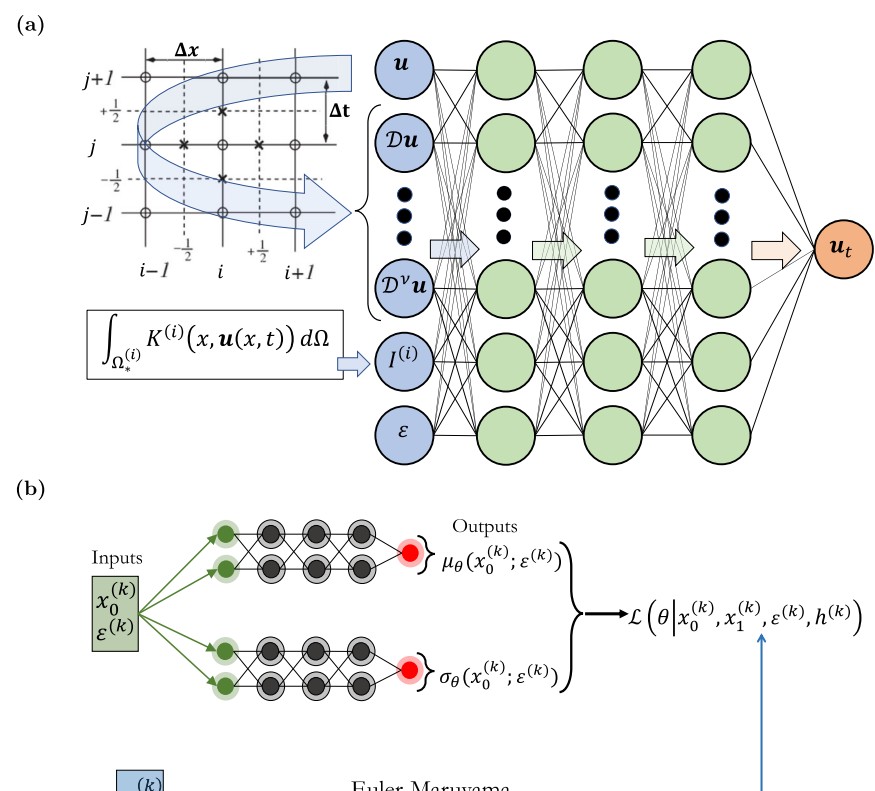

**Fig. 5 | Schematic of the neural networks used for constructive machine learning assisted surrogates. a** Feedforward Neural Network (FNN). the input is constructed by convolution operations, i.e., a combination of sliding Finite Difference (FD) stencils, and, integral operators, for learning mesoscopic models in the form of IPDEs (Eq. (10)); the inputs to the RHS of the IPDE are the features in Eq. (13). **b** A schematic of the neural network architecture, inspired by numerical stochastic integrators, used to construct macroscopic models in the form of mean-field SDEs.

stochasticity, a natural first choice is the construction of a mean-field macroscopic SDE. Here, we construct surrogate models via machine learning at both these distinct coarse-grained scales.

## Learning mesoscopic IPDEs via neural networks

As we have discussed in the introduction, the identification of evolution operators of spatio-temporal dynamics using machine learning tools, including deep learning and Gaussian processes, represents a well-established field of research. The main assumption here is that the emergent dynamics of the complex system under study on a domain $\Omega \times [t_0, t_{end}] \subseteq \mathbb{R}^d \times \mathbb{R}$ can be modeled by a system of say $m$ IPDEs in the form of:

$$
\frac{\partial u^{(i)}(\mathbf{x},t)}{\partial t} \equiv u_t^{(i)} = F^{(i)}\Big(\mathbf{x}, \mathbf{u}, \mathcal{D}\mathbf{u}, \mathcal{D}^2\mathbf{u}, \ldots,
$$
$$
\ldots, \mathcal{D}^{\boldsymbol{\nu}}\mathbf{u}, I_1^{(i)}(\mathbf{u}), I_2^{(i)}(\mathbf{u}), \ldots, \boldsymbol{\varepsilon}\Big),
$$
$$
(\mathbf{x},t) \in \Omega \times [t_0, t_{end}], \quad i = 1, 2, \ldots, m,
$$

$$(10)$$

where $F^{(i)}, i = 1, 2, \ldots m$ are $m$ nonlinear integro-differential operators; $\mathbf{u}(\mathbf{x}, t) = [u^{(1)}(\mathbf{x}, t), \ldots, u^{(m)}(\mathbf{x}, t)]$ is the vector containing the spatio-temporal fields, $\mathcal{D}^{\boldsymbol{\nu}}\mathbf{u}(\mathbf{x},t)$ is the generic multi-index $\boldsymbol{\nu}$-th order spatial derivative at time $t$:

$$
\mathcal{D}^{\boldsymbol{\nu}}\mathbf{u}(\mathbf{x},t) := \left\{ \frac{\partial^{|\boldsymbol{\nu}|}\mathbf{u}(\mathbf{x},t)}{\partial x_1^{\nu_1} \cdots \partial x_d^{\nu_d}}, \nu_1, \ldots, \nu_d \geq 0 \right\},
$$
$$
\text{where } |\boldsymbol{\nu}| = \nu_1 + \nu_2 + \cdots + \nu_d,
$$

$$(11)$$

$I_1^{(i)}, I_2^{(i)}, \ldots$ are a collection of integral features on subdomains $\Omega_1^{(i)}, \Omega_2^{(i)}, \cdots \subseteq \Omega$:

$$
I_j^{(i)}(\mathbf{u}) = \int_{\Omega_j^{(i)}} K_j^{(i)}(\mathbf{x}, \mathbf{u}(\mathbf{x}, t)) d\Omega, j = 1, 2, \ldots;
$$

$$(12)$$

$K_j^{(i)} : \mathbb{R}^d \times \mathbb{R}^m \mapsto \mathbb{R}^d$ are nonlinear maps and $\boldsymbol{\varepsilon} \in \mathbb{R}^p$ denotes the (bifurcation) parameters of the system. The right-hand-side of the $i$-th IPDE depends on say, a number of $\gamma^{(i)}$ variables and on bifurcation parameters from the set of features:

$$
\mathcal{S}^{(i)} = \Big\{ \mathbf{x}, \mathbf{u}(\mathbf{x}, t), \mathcal{D}\mathbf{u}(\mathbf{x}, t), \mathcal{D}^2\mathbf{u}(\mathbf{x}, t), \ldots,
$$
$$
\mathcal{D}^{\boldsymbol{\nu}}\mathbf{u}(\mathbf{x}, t), I_1^{(i)}, I_2^{(i)}, \ldots, \boldsymbol{\varepsilon} \Big\}.
$$

$$(13)$$

At each spatial point $\mathbf{x}_q, q = 1, 2, \ldots, M$ and time instant $t_s, s = 1, 2, \ldots, N$, a single sample point (an observation) in the set $\mathcal{S}^{(i)}$ for the $i$-th IPDE can be described by a vector $Z_j^{(i)} \equiv Z_{(q,s)}^{(i)} \in \mathbb{R}^{\gamma^{(i)}}$, with $j = q + (s-1)M$. Here, we assume that such mesoscopic IPDEs in principle exist, but they are not available in closed-form. Henceforth, we aim to learn the macroscopic laws by employing a Feedforward Neural Network (FNN), in which the effective input layer is constructed by a finite stencil (sliding over the computational domain), mimicking convolutional operations where the applied "filter" involves values of our field variable(s) $u^{(i)}$ on the stencil, and returns features $Z_j^{(i)} \in \mathcal{S}^{(i)}$ of these variables at the stencil center-point, i.e., spatial derivatives as well as (local or global) integrals (see Fig. 5a for a schematic).

The FNN is fed with the sample points $Z_j^{(i)} \in \mathbb{R}^{\gamma^{(i)}}$ and its output is an approximation of the time derivative $\mathbf{u}_t$. In sections D2 and D3 of the Supplementary Information (SI) we describe two different approaches (gradient descent and random projections) to train such a FNN. For alternative operator-learning approximation methods, see e.g., ref. 74 and ref. 51.

## Remark on learning mean-field ODEs

The proposed framework can be also applied for the simpler task of learning a system of $m$ ODEs in terms of the $m$ variables $\mathbf{u} = (u^{(1)}, u^{(2)}, ..., u^{(m)})$, thus learning $i = 1, ..., m$ functions:

$$\frac{du^{(i)}(t)}{dt} = F^{(i)}(\mathbf{u}, \mathbf{p}), t \in [t_0, t_{end}], \quad (14)$$

where $\mathbf{p} \in \mathbb{R}^k$ denotes the parameter vector.

## Feature selection with automatic relevance determination

Training the FNN with the "full" set of inputs $\mathcal{S}^{(i)} \subseteq \mathbb{R}^{\gamma^{(i)}}$, described in Eq. (13), consisting of all local mean field values as well as all their coarse-scale spatial derivatives (up to some order $\nu$) is prohibitive due to the *curse of dimensionality*. Therefore, one important task for the training of the FNN is to extract a few "relevant"/dominant variable combinations. Towards this aim, we used Automatic Relevance Determination (ARD) in the framework of Gaussian processes regression (GPR)[29]. The approach assumes that the collection of all observations $\mathbf{Z}^{(i)} = (Z_1^{(i)}, Z_2^{(i)}, \dots, Z_{MN}^{(i)})$, of the features $z_l \in \mathcal{S}^{(i)}$, are a set of random variables whose finite collections have a multivariate Gaussian distribution with an unknown mean (usually set to zero) and an unknown covariance matrix $K$. This covariance matrix is commonly formulated by a Euclidean distance-based kernel function $k$ in the input space, whose hyperparameters are optimized based on the training data. Here, we employ a radial basis kernel function (RBF), which is the default kernel function in Gaussian process regression, with ARD:

$$K_{jh}^{(i)} = k(Z_j^{(i)}, Z_h^{(i)}, \boldsymbol{\theta}^{(i)}) = \theta_0^{(i)} \exp\left(-\frac{1}{2} \sum_{l=1}^{\gamma^{(i)}} \frac{z_{l,j} - z_{l,h}}{\theta_l^{(i)}}\right); \quad (15)$$

$\boldsymbol{\theta}^{(i)} = [\theta_0^{(i)}, \theta_1^{(i)}, \dots, \theta_{\gamma^{(i)}}^{(i)}]$ are a $(\gamma^{(i)} + 1)$-dimensional vector of hyperparameters. The optimal hyperparameter set $\tilde{\boldsymbol{\theta}}^{(i)}$ can be obtained by minimizing a negative log marginal likelihood over the training data set $(\mathbf{Z}^{(i)}, \mathbf{Y}^{(i)})$, with inputs the observation $\mathbf{Z}^{(i)}$ of the set $\mathcal{S}^{(i)}$ and corresponding desired output given by the observation $\mathbf{Y}^{(i)}$ of the time derivative $u_t^{(i)}$:

$$\tilde{\boldsymbol{\theta}}^{(i)} = \arg\min_{\boldsymbol{\theta}^{(i)}} - \log p(\mathbf{Y}^{(i)} | \mathbf{Z}^{(i)}, \boldsymbol{\theta}^{(i)}). \quad (16)$$

As can be seen in Eq. (15), a large value of $\theta_l$ nullifies the difference between target function values along the $l$-th dimension, allowing us to designate the corresponding $z_l$ feature as "insignificant". Practically, in order to build a reduced input data domain, we define the normalized effective *relevance weights* $W_r^{(i)}(\cdot)$ of each feature input $z_l \in \mathcal{S}^{(i)}$, by taking:

$$\bar{W}_r^{(i)}(z_l) = exp(-\tilde{\theta}_l^{(i)}), W_r^{(i)}(z_l) = \frac{\bar{W}_r^{(i)}(z_l)}{\sum_l \bar{W}_r^{(i)}(z_l)}. \quad (17)$$

Thus, we define a small tolerance *tol* in order to disregard the components such that $W_r^{(i)}(z_l) < tol$. The remaining selected features $(W_r^{(i)}(z_l) \geq tol)$ can still successfully (for all practical purposes) parametrize the approximation of the right-hand-side of the underlying IPDE.

## Macroscopic mean-field SDEs via neural networks

Here, we present our approach for the construction of embedded surrogate models in the form of mean-field SDEs. Under the assumption that we are close (in phase- and parameter space) to a previously located tipping point, we can reasonably assume that the effective dimensionality of the dynamics can be reduced to the corresponding normal form. We already have some qualitative insight on the type of the tipping point, based for example on the numerical bifurcation calculations that located it (e.g., for the financial ABM from the analytical FP IPDE, from our surrogate IPDE, or from the EF analysis[52,56], while for the epidemic ABM from the EF analysis in ref. 59. For both these two particular problem, we have found that the tipping point corresponds to a saddle-node bifurcation.

Given the nature of the bifurcation (and the single variable corresponding normal form) we identify a one-dimensional SDE, driven by a Wiener process, from data. We note that learning higher-order such SDEs, or SDEs based on the more general Lévy process and the Ornstein-Uhlenbeck process[75], is straightforward.

For a diffusion process with drift, say $X_t = \{x_t, t > 0\}$, the drift, $\mu(x_t)$ and diffusivity $\sigma^2(x_t)$ coefficients over an infinitesimally small-time interval $dt$, are given by:

$$\begin{aligned} \mu(x_t) &= \lim_{\delta t \to 0} \frac{1}{\delta t} \mathbb{E}(\delta x_t | X_t = x_t), \\ \sigma^2(x_t) &= \lim_{\delta t \to 0} \frac{1}{\delta t} \mathbb{E}(\delta x_t^2 | X_t = x_t), \end{aligned} \quad (18)$$

where, $\delta x_t = x_{t+\delta t} - x_t$.

The 1D SDE driven by a Wiener process $W_t$ reads:

$$dx_t = \mu(x_t; \varepsilon)dt + \sigma(x_t; \varepsilon)dW_t. \quad (19)$$

Here, for simplicity, we assume that the one-dimensional parameter $\varepsilon$, enters into the dynamics, via the drift and diffusivity coefficients. Note that the parameter $\varepsilon$ can be either the parameter $g$ introduced in Eq. (3) or the parameter $\lambda$ for the epidemic ABM (representing the probability that a susceptible individual may get infected). Our goal is to identify the functional form of the drift $\mu(x, \varepsilon)$ and the diffusivity $\sigma(x, \varepsilon)$ given noisy data close to the tipping point via machine learning. For the training, the data might be collected from either long-time trajectories or short bursts initialized at scattered snapshots, as in the EF framework. These trajectories form our data set of input-output pairs of discrete-time maps. A data point in the collected data set can be written as $(x_0^{(k)}, h^{(k)}, x_1^{(k)} \varepsilon^{(k)})$, where $x_0^{(k)}$ and $x_1^{(k)}$ measures two consecutive states at $t_0^{(k)}$ and $t_1^{(k)}$ with (small enough) time step $h^{(k)} = t_1^{(k)} - t_0^{(k)}$ and $\varepsilon^{(k)}$ is the parameter value for this pair. Based on the above formulation, going to $x_1^{(k)}$ from $x_0^{(k)}$ by:

$$x_1^{(k)} = x_0^{(k)} + \int_{t_0^{(k)}}^{t_1^{(k)}} \mu(x_t; \varepsilon^{(k)})dt + \int_{t_0^{(k)}}^{t_1^{(k)}} \sigma(x_t; \varepsilon^{(k)})dW_t. \quad (20)$$

Here, for the numerical integration of the above equation to get stochastic realizations of $x_1^{(k)}$, we assume that the Euler-Maruyama numerical scheme can be used, reading:

$$x_1^{(k)} \approx x_0^{(k)} + h^{(k)}\mu(x_0^{(k)}; \varepsilon^{(k)}) + \sigma(x_0^{(k)}; \varepsilon^{(k)})\Delta W^{(k)}, \quad (21)$$

where $\Delta W^{(k)} = W_{t_1^{(k)}} - W_{t_0^{(k)}} \in \mathbb{R}$ is a one-dimensional random variable, normally distributed with expected value zero and variance $h^{(k)}$.

Considering the point $x_1^{(k)}$ as a realization of a random variable $X_1$, conditioned on $x_0^{(k)}$ and $h^{(k)}$, drawn by a Gaussian distribution of the

form:

$$p_{X_1}(x_1^{(k)}) = \mathbb{P}\left(X_1 = x_1^{(k)} | X_0 = x_0^{(k)}, h^{(k)}\right) \sim$$
$$\sim \mathcal{N}\left(x_0^{(k)} + h^{(k)}\mu(x_0^{(k)}; \varepsilon^{(k)}), h^{(k)}\sigma(x_0^{(k)}; \varepsilon^{(k)})^2\right), \tag{22}$$

we approximate the drift $\mu(x_0^{(k)}; \varepsilon^{(k)})$ and diffusivity $\sigma(x_0^{(k)}; \varepsilon^{(k)})$ functions by simultaneously training two neural networks, denoted as $\mu_\theta$ and $\sigma_\theta$, respectively. This training process involves minimizing the loss function:

$$\mathcal{L}(\theta | x_0^{(k)}, x_1^{(k)}, h^{(k)}) :=$$
$$= \sum_k \frac{(x_1^{(k)} - x_0^{(k)} - h^{(k)}\mu_\theta(x_0^{(k)}; \varepsilon^{(k)}))^2}{h^{(k)}\sigma_\theta(x_0^{(k)}; \varepsilon^{(k)})^2} + \tag{23}$$
$$+ \log|h^{(k)}\sigma_\theta(x_0^{(k)}; \varepsilon^{(k)})^2|.$$

which is derived in order to maximize the log-likelihood of the data and where $\theta$ denotes the trainable parameters (e.g., weights and biases of the neural networks $\mu_\theta$ and $\sigma_\theta$). A schematic of the Neural Network, based on Euler-Maruyama, is shown in Fig. 5b.

### Locating tipping points via our surrogate models
In order to locate the tipping point, based on either the mesoscopic IPDE or the embedded mean-field 1D SDE model, we construct the corresponding bifurcation diagram in its neighborhood, using pseudo-arc-length continuation as implemented in numerical bifurcation packages. For the identified SDE, we used its deterministic part, i.e., the drift term, to perform continuation. The required Jacobian of the activation functions of the neural network is computed by symbolic differentiation. Note that this is just a validation step (we already know the location and nature of the tipping point).

### Rare-event analysis/UQ of catastrophic shifts
Given a sample space $\Omega$, an index set of times $T = \{0, 1, 2, \dots\}$ and a state space $S$, the *first passage time*, also known as *mean exit time* or *mean escape time*, of a stochastic process $x_t: \Omega \times T \mapsto S$ on a measurable subset $A \subseteq S$ is a random variable which can be defined as

$$\tau(\omega) := \inf\{t \in T | x_t(\omega) \in S \setminus A\}, \tag{24}$$

where $\omega$ is a sample out of the space $\Omega$. One can define the mean escape time from $A$, which works as the expectation of $\tau(\omega)$:

$$\langle\tau\rangle := \mathbb{E}[\tau(\omega)]. \tag{25}$$

For a $n$-dimensional stochastic process, as it is the ABM under study, $S$ is typically set to be $\mathbb{R}^n$, and $A$ is usually a bounded subset of $\mathbb{R}^n$. In the case of our local 1D SDE model, the subset $A$ reduces to an open interval $(a, b)$, with the initial condition of this stochastic process $x_0$ also chosen in this interval.

We discuss two ways for quantifying the uncertainty of the occurrence of those rare-events. The first, presented in the main text, involves direct computational "cheap" temporal simulations of the 1D SDE, where one gets an empirical probability distribution; the second, presented only in section F4 of the Supplementary Information (SI), is a closed-form expression, based on statistical mechanics[76], for the mean escape time (assuming an exponential distribution of escape times).

### Computation of escape times based on temporal simulations of the 1D SDE
We perform numerical integration of multiple stochastic trajectories of the SDE to obtain an estimation of the desired mean escape time.

The algorithm for estimating the escape times of a one-dimensional stochastic process with initial condition $x_0$ on the interval $(a, b)$ can be described as follows:

1. Given a fixed time step $h > 0$, we perform $i + 1$ numerical integration steps of the SDE until a stopping condition, i.e., $x_t$ exits, for the first time $t_{i+1}$, the interval at $a$ or $b$.
2. Record the time $\tau = t_i$, which corresponds to a realization of the escape time.
3. Repeat the above steps 1–2 for $K$ iterations, and each time collect the observed escape time.
4. Compute the statistical mean value of the collected escape times that corresponds to an approximation of the mean escape time.

In our case, the initial condition $x_0$ was set equal to the stable steady state, and the termination condition $a$ or $b$ at which we consider the dynamics escaped/exploded.

### Reporting summary
Further information on research design is available in the Nature Portfolio Reporting Summary linked to this article.

## Data availability
Data used in this work are publicly available in the GitLab repository https://gitlab.com/nicolasevangelou/agent_based.

## Code availability
The code used to produce the findings of this study are publicly available in the GitLab repository https://gitlab.com/nicolasevangelou/agent_based.

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

## Acknowledgements

The US AFOSR FA9550-21-0317 and the US Department of Energy SA22-0052-S001 partially supported I.G.K. and N.E. The PNRR MUR, projects PE0000013-Future Artificial Intelligence Research-FAIR & CN0000013 CN HPC - National Centre for HPC, Big Data and Quantum Computing, Gruppo Nazionale Calcolo Scientifico-Istituto Nazionale di Alta Mate-matica (GNCS-INdAM) partially supported C.S. "la Caixa" Foundation Fellowship (ID100010434), code LCF/BQ/AA19/11720048 supported C.P.M.L.

## Author contributions

G.F.: Data curation, formal analysis, investigation, methodology, soft-ware, validation, visualization, writing—original draft, writing—review & editing. N.E.: Data curation, formal analysis, investigation, methodology, software, validation, visualization, writing—original draft, writing—review & editing. T.C.: Data curation, methodology, software, writing—original draft. J.M.B.R.: Data curation, formal analysis, investigation, methodology, software, writing—review & editing. C.P.M.L.: Data cura-tion, investigation, methodology, software. C.S.: Conceptualization, formal analysis, supervision, investigation, methodology, software, validation, writing—review & editing. I.G.K.: Conceptualization, formal analysis, supervision, investigation, methodology, validation, writing—review & editing.

## Competing interests

The authors declare no competing interests.
