## [Peer Review File · Nature Communications]

REVIEWER COMMENTS

Reviewer #1 (Remarks to the Author):

The paper describes a framework for detecting tipping points in complex systems and creating models at various scales to perform different types of analyses that correspond to system descriptions in different scales.

I find the concepts, that I think the paper aims to present, very interesting and probably worthy of publication. However, the paper requires a major rewriting in order to become better accessible and to clearly demonstrate its contributions.

Reading the paper requires flipping a lot between Appendices and the main text that makes for a confusing reading. For example a parameter g that is critical in understanding various plots has to be dug out from the Appendix to find its definition. Several concepts are described in a confusing manner and it is unclear what has been previously known and what are the novel contributions of this article.

The work and conclusions presented in the paper are based in a microscopic description of a simplified agent based model simulation of financial markets. I miss some sort of validation or relevant verification of any kind for this model.

Last but not least, while I find the ideas very useful, having them demonstrated with one more micro/macro models besides the one herein, would have strengthened tremendously the paper and would have better justified its title.

In summary, this is a very interesting paper that needs a thorough rewriting to become more accessible. I would be happy to review a revised version of this article.

Reviewer #2 (Remarks to the Author):

Summary: The authors propose a framework for extracting mesoscopic and macroscopic characterizations in the form of, respectively, integro- and stochastic differential equations from large-scale agent-based models. The resulting reduced models are then used to find and characterize tipping points (where the system exhibits drastic dynamical changes) and the escape times (at which these state changes are observed). The general framework is implemented for an agent-based model of a financial market that exhibits "financial bubbles" as tipping points depending on the level of mimesis utilized by traders.

Assessment: I found this paper stimulating to read. While the underlying methodologies are well known, and some of the results have been published previously, the proposed framework is interesting, and the results noteworthy. The work reported here should be of interest to the broader community, and the work supports the conclusion drawn in the paper.

This being said, the central question used by the authors to frame their paper is how tipping points and rare-event probabilities can be inferred in complex large-scale systems. This is a very important, challenging, and timely question. My concern is that it is not clear to what extent the authors' results address this question. The reduced-order models are extracted from the underlying agent-based model by simulating the agent-based model for a range of values of the parameter g , including values at which the system is already beyond tipping. It is then not clear how useful this approach is in practice to predict the onset of tipping without actually being in the regime where system already exhibits the drastic changes that we want to predict.

I therefore ask the authors to be more clear about how they think about their results: what is the scope of the proposed framework, what can we realistically expect it to do, and what are its limitations?

Other comments:

Title: English is not my first language, and I never learned "old English". It took me some time to understand what "makyth" refers to (and the paragraph in column 2 on p2 did not help), which distracted me from focusing on the content of the paper. I suggest to either explain this word on the title page or change it to "Tasks make models" and explain the context later.

Abbreviations: The authors use many abbreviations in their manuscript. While I understand the reason, and while some of the abbreviations are either well known or at least intuitive, others are hard to remember (DM=direct message?), especially as they are sometimes used several pages after they have

been introduced. Please be kind to your readers and remind them of these abbreviations or, ideally, avoid them -- trying to remember each abbreviation distracted me from the content.

Financial market with mimesis: Please add a brief explanation of what mimesis refers to in this context.

Figure 1: It is difficult to identify the differences between the graphs and color plots of the PDFs in panels (a) and (b). It might be easiest to just provide the color (heat) maps plus the graphs for the PDFs near the end of the simulation, which contain all the information you likely need to convey.

Meaning of the state variable X : The descriptions in the left and right column for the meaning of X on p3 seem contradictory: on the left, $X=1$ refers to selling stock (which aligns with my interpretation of what a financial bubble is), while on the right the interpretation is the opposite ($X=1$ refers to buying, which is not something I associate with a bubble).

Figure 5: Please make the colors of the histograms consistent across panels (e) and (f) (blue in (e) corresponds to red in (f)). Alternatively, mention the discussion of these graphs from column 2 on p9 in the caption of the figure to explain why readers should compare the two red histograms and not the ones for the same value of g (also, why are these more comparable?).

Task-specific Machine Learning Surrogates for Tipping Points of Agent-based Models

Gianluca Fiabiani, Nikolaos Evangelou, Tianqi Cui, Juan M. Bello-Rivas, Cristina P. Martin-Linares, Constantinos Siettos*, Ioannis G. Kevrekidis*

* Corresponding Authors: Ioannis G. Kevrekidis — Email: yannisk@jhu.edu, Constantinos Siettos — Email: constantinos.siettos@unina.it

We would like to thank both referees for their time and effort spent for reviewing the manuscript and for their positive and constructive comments. We really appreciate it. In the following, we respond point by point to the comments of the referees; our responses are written using blue color.

Reviewer 1

The paper describes a framework for detecting tipping points in complex systems and creating models at various scales to perform different types of analyses that correspond to system descriptions in different scales. I find the concepts, that I think the paper aims to present, very interesting and probably worthy of publication. However, the paper requires a major rewriting in order to become better accessible and to clearly demonstrate its contributions. Reading the paper requires flipping a lot between Appendices and the main text that makes for a confusing reading. For example a parameter g that is critical in understanding various plots has to be dug out from the Appendix to find its definition. Several concepts are described in a confusing manner and it is unclear what has been previously known and what are the novel contributions of this article.

The reviewer is right. We have now restructured the manuscript as suggested, so that the presentation of the methodology, benchmark problems and novel contributions appear in a more reasonable order, and detailed enough for clarity. We have included in the main text all the necessary information about the models (and the parameters). We have also included the analytical model formulas (e.g. Fokker-Planck-type equations) in the main text. Finally, we hope the current version of the paper illustrates our contributions more clearly to the reader.

The work and conclusions presented in the paper are based in a microscopic description of a simplified agent based model simulation of financial markets. I miss some sort of validation or relevant verification of any kind for this model.

To address this, we have now added the following paragraph in the section where we describe the ABM financial model:

ABMs allow for the creation of digital twins for financial markets, thus offering a valuable tool in our arsenal for explaining out-of-equilibrium phenomena such as “bubbles” and market crashes (Farmer & Foley, Nature, 2009) that emerge mainly due to positive feedback mechanisms of imitation and herding of investors (Sornette, Journal of Economic Interaction and Coordination, 2008) (see for example the Santa Fe artificial stock market (LeBaronm Physica A, 2002), and the EURACE ABM for modelling the European economy (Deissenberg, Applied mathematics and computation, 2008). While the practical application of ABMs for providing predictions about real-world financial instabilities remains an ongoing area of research, they are fundamental to shed light into the mechanisms that lead to such crises (Farmer & Foley, Nature, 2009)”.

Last but not least, while I find the ideas very useful, having them demonstrated with one more micro/macro models besides the one herein, would have strengthened tremendously the paper and would have better justified its title.

We have now added a second ABM model, describing epidemic dynamics over a (Erdős-Rényi) social network. For this ABM a mean field SIR can be derived analytically at the limit of infinite number of agents limit, and homogeneous social network. However, such model fails to identify the tipping point accurately.

We showcase the proposed approach in this new case study by learning: (a) an accurate data-driven mean-field level effective model (two coupled ODEs); and (b) a one-dimensional, parameter dependent effective SDE (close to the tipping point). Both models accurately capture the tipping point. Yet the effective SDE is the appropriate model type for performing rare event computations.

In summary, this is a very interesting paper that needs a thorough rewriting to become more accessible. I would be happy to review a revised version of this article.

Thank you for the comment. We have restructured the paper and included an additional example as you suggested. We hope that you find the revised paper more accessible, and hopefully acceptable for publication.

Reviewer 2

The authors propose a framework for extracting mesoscopic and macroscopic characterizations in the form of, respectively, integro- and stochastic differential equations from large-scale agent-based models. The resulting reduced models are then used to find and characterize tipping points (where the system exhibits drastic dynamical changes) and the escape times (at which these state changes are observed). The general framework is implemented for an agent-based model of a financial market that exhibits “financial bubbles” as tipping points depending on the level of mimesis utilized by traders.

I found this paper stimulating to read. While the underlying methodologies are well known, and some of the results have been published previously, the proposed framework is interesting, and the results noteworthy. The work reported here should be of interest to the broader community, and the work supports the conclusion drawn in the paper.

This being said, the central question used by the authors to frame their paper is how tipping points and rare-event probabilities can be inferred in complex large-scale systems. This is a very important, challenging, and timely question. My concern is that it is not clear to what extent the authors’ results address this question. The reduced-order models are extracted from the underlying agent-based model by simulating the agent-based model for a range of values of the parameter g , including values at which the system is already beyond tipping. It is then not clear how useful this approach is in practice to predict the onset of tipping without actually being in the regime where system already exhibits the drastic changes that we want to predict. I therefore ask the authors to be more clear about how they think about their results: what is the scope of the proposed framework, what can we realistically expect it to do, and what are its limitations?

This question was also posed by the first reviewer for our agent-based model of a financial market. Our work is not about the construction and validation of early warning systems based on real-world data. Our main target is to show how via Machine Learning, one can systematically deal with the “curse of dimensionality” when trying to analyse the emergent behaviour of ABMs (a.k.a. “digital twins”). The particular task of interest is the construction of different types of reduced-order models for understanding, analysing the mechanisms governing the emergence of– tipping points, and quantifying the probability of the occurrence of rare events close to them. This is an open and challenging question associated with the “curse of dimensionality” when trying to learn appropriate surrogate models for large-scale ABMs.

To better clarify the scope of the current work, we have now added (1) a new paragraph in the introduction, for the challenge of dealing with the curse of dimensionality when exploring ABMS (citing key papers on this matter); (2) a new paragraph in the section which describes the ABM financial model, discussing (and in the process, citing key works in the field) the importance of analysing in an efficient manner large scale ABMs (3) a paragraph in the discussion section, clarifying further the scope and importance of the current work. Let us just quickly mention here in passing that one needs a few full ABM simulations in the neighborhood of the tipping point to inform the effective single SDE; and this SDE can then be used in many, many individual realizations to estimate escape time statistics - we do not just use to “repeat” the full simulations.

We have also significantly revised, the Abstract, Introduction and Discussion sections, in order to clarify better the pros and cons of the proposed approach.

Other Comments:

English is not my first language, and I never learned “old English”. It took me some time to understand what “makyth” refers to (and the paragraph in column 2 on p2 did not help), which distracted me from focusing on the content of the paper. I suggest to either explain this word on the title page or change it to “Tasks make models” and explain the context later.

We have now changed the title to “Task-specific Machine Learning Surrogates for Tipping Points of Agent-based Models”. However, just for your information, Rutherford Aris, the advisor of one of the corresponding authors, who was British, once wrote a paper entitled “Manners Makyth Modelers” paraphrasing William Wykeham’s statement “Manners Makyth Man”. In our case, this new paraphrasing was both appropriate for the idea of the paper (different computational tasks require different type models) and, at the same time, a tribute to R. Aris. Still, as suggested, we changed the title and mention something about it in the main text.

The authors use many abbreviations in their manuscript. While I understand the reason, and while some of the abbreviations are either well known or at least intuitive, others are hard to remember (DM=direct message?), especially as they are sometimes used several pages after they have been introduced. Please be kind to your readers and remind them of these abbreviations or, ideally, avoid them – trying to remember each abbreviation distracted me from the content.

Thank you for pointing this out. We have restructured the paper and tried to improve the readability of it. We also changed the abbreviation of Diffusion Maps to DMaps.

Financial market with mimesis: Please add a brief explanation of what mimesis refers to in this context.

We included a footnote in the paper that describes that mimesis refers to “the traders in this ABM tend to imitate the behavior of other traders, because of social conformity or subtle psychological pressure to align their behavior with that of other agents (their peers)”.

Figure 1: It is difficult to identify the differences between the graphs and color plots of the PDFs in panels (a) and (b). It might be easiest to just provide the color (heat) maps plus the graphs for the PDFs near the end of the simulation, which contain all the information you likely need to convey.

We appreciate your suggestion. We attempted to illustrate the 2D contour plot, but it shows less informative compared to the surface plot and corresponding insets that we used to show the blowing up. To better understand the difference, please refer to the provided figure here below (to be contrasted with the current Figure 2. Hence, we decided to keep the original figure - we hope you will not disagree):

Meaning of the state variable X: The descriptions in the left and right column for the meaning of X on p3 seem contradictory: on the left, X=1 refers to selling stock (which aligns with my interpretation of what a financial bubble is), while on the right the interpretation is the opposite (X=1 refers to buying, which is not something I associate with a bubble).

We now explain in detail, and more clear the ABM model, its variables and parameters in the main text. The state variable X indicates the preference state of each agent. The definitions of $X = 1$ as buying and $X = -1$ as selling were used in the original construction of the model in the paper by Omurtag and Sirovich 2006. We kept the same notation for consistency. Regarding the part of the question about a “financial bubble”: an economic bubble is caused by a high demand for a particular asset, an over-optimistic sentiment. This leads to increased investments and demand for assets in that area, which leads to a rapid escalation of the prices of the assets which greatly exceed their intrinsic valuation. In the ABM model, when $g > g^$ we find a regime where everybody buys. This is manifested in the model as a singularity/blowup in the evolution*

of the IPDE. In the ABM model, the price of the stock is not modelled but just the demand and offer (the buying and selling rates). . The ABM is capable of producing both “bubbles” (shown in the paper, where everybody “rushes to buy”) as well as “crashes”- where everybody “rushes to sell”, which aligns more with what you describe. We have now added a paragraph citing key papers explaining that bubbles emerge mainly due to positive feedback mechanisms of imitation and herding of investors that lead to an escalating increase of the demand.

Figure 5: Please make the colors of the histograms consistent across panels (e) and (f) (blue in (e) corresponds to red in (f)). Alternatively, mention the discussion of these graphs from column 2 on p9 in the caption of the figure to explain why readers should compare the two red histograms and not the ones for the same value of g (also, why are these more comparable?).

Thank you for your comment. We implemented the suggested change in the colors of those figures.

REVIEWERS' COMMENTS

Reviewer #1 (Remarks to the Author):

The authors have addressed my concerns. I enjoyed reading the revised version and I recommend publication.

Reviewer #2 (Remarks to the Author):

In the revised version, the authors addressed all of my suggestions and concerns adequately. The addition of a second model system demonstrates the broader applicability of the proposed framework, and the content is now also more accessible and easier to follow. The authors clarified the scope of their work, which is focused on deriving, in a data-driven way, different reduced models of large-scale systems (here primarily large agent-based models) to elucidate the nature and the mechanisms behind tipping transitions in these systems.

As I wrote in my original review, the paper is stimulating to read, and I find the concepts and ideas presented here very interesting. With the revisions, the results are clearly communicated, and I believe that this paper will have impact in the community.

Reviewer #2 (Remarks on code availability):

I could not open the URL provided by the authors: I get a 404 Page not found error.

Task-Oriented Machine Learning Surrogates for Tipping Points of Agent-based Models

Gianluca Fabiani, Nikolaos Evangelou, Tianqi Cui, Juan M. Bello-Rivas, Cristina P. Martin-Linares, Constantinos Siettos*, Ioannis G. Kevrekidis*

* Corresponding Authors: Ioannis G. Kevrekidis — Email: yannisk@jhu.edu, Constantinos Siettos — Email: constantinos.siettos@unina.it

We would like to thank both referees for their time and effort spent for reviewing the manuscript and for their positive responses. We really appreciate it. In the following, we respond point by point to the comments of the referees; our responses are written using blue color.

Reviewer 1

The authors have addressed my concerns. I enjoyed reading the revised version and I recommend publication.

Thank you so much for your kind words and once more for taking the time to review our manuscript.

Reviewer 2

In the revised version, the authors addressed all of my suggestions and concerns adequately. The addition of a second model system demonstrates the broader applicability of the proposed framework, and the content is now also more accessible and easier to follow. The authors clarified the scope of their work, which is focused on deriving, in a data-driven way, different reduced models of large-scale systems (here primarily large agent-based models) to elucidate the nature and the mechanisms behind tipping transitions in these systems.

As I wrote in my original review, the paper is stimulating to read, and I find the concepts and ideas presented here very interesting. With the revisions, the results are clearly communicated, and I believe that this paper will have impact in the community.

We would like to thank you once more for your time on reviewing our work.

Reviewer 2 (Remarks on code availability): I could not open the URL provided by the authors: I get a 404 Page not found error.

We have now made the repository public and should be accessible in the link https://gitlab.com/nicolasevangelou/agent_based